# Microbial Solution of Growth-Promoting Bacteria Sprayed on Monoammonium Phosphate for Soybean and Corn Production

**Cristiane Prezotto Silveira** [1][ID], **Fernando Dini Andreote** [1], **Risely Ferraz-Almeida** [1][ID], **Jardelcio Carvalho** [2], **John Gorsuch** [2] and **Rafael Otto** [1,*][ID]

1   Department of Soil Science, Luiz de Queiroz College of Agriculture, University of São Paulo, 11 Pádua Dias Ave., Piracicaba 13418-900, Brazil
2   BiOWiSH Technologies, 2717 Erie Ave., Cincinnati, OH 45208, USA
*   Correspondence: rotto@usp.br

**Abstract:** Common fertilizers present a low use efficiency caused by nutrient losses (e.g., through leaching, volatilization, adsorption, and precipitation in solution as well as through microbial reduction and immobilization) that create a significant limiting factor in crop production. Inoculation with Plant Growth-Promoting Bacteria (PGPB) is presented as an alternative to increasing fertilizer efficiency. The goal of the study was to test the hypothesis that PGPB (solution with *Bacillus subtilis*, *Bacillus amyloliquefaciens*, *Bacillus licheniformis*, and *Bacillus pumilus*) can be a strategy to increase the monoammonium phosphate (MAP) efficiency, root growth, and nutrient assimilation of soybean and corn cultivated in arenosol and oxisol. A greenhouse study was developed with the rates of PGPB (rates: 0, 1, 1.33, and 1.66–2.0 L per ton of fertilizer) sprayed on MAP and applied in an arenosol and oxisol cultivated with soybean and corn. Results showed that in both soils and crops, there was a variation in soil biological activity during the experiment. On day 45, PGPB + MAP promoted the beta-glucosidase and ammonium-oxidizing microorganism activities in the arenosol. The PGPB + MAP increased crop root growth in both soils and crops. Plant dry matter was associated with the phosphorous content in the soil, indicating that the phosphorous applied was absorbed by the plants, consequently resulting in a higher accumulation in the plant. Based on the results, the conclusion is that PGPB + MAP increases the growth and phosphorous accumulation of soybean and corn cultivated in the arenosol and oxisol, with a direct effect on crop rooting.

**Keywords:** fertilizers; *Bacillus* sp.; soil enzymes; soil biological activity; microorganisms; *Glycine max*; *Zea mays*

## 1. Introduction

Phosphorus (P) is essential and a limiting nutrient for crop production, playing in some physiological and biochemical processes in plants, such as the photosynthesis process, nucleic acid synthesis, and enzyme activity regulation. However, P nutrition is a major limiting factor for crop production in many soils [1,2]. In soil, P can be found in inorganic and organic forms with an intense formation and transformation, which can be affected by many soil factors, including physical and chemical processes, roots, microorganisms, and soil management [3].

In soils, the phosphorus available for plant absorption generally is low due to the phosphorus adsorption on the positively charged surface of minerals (oxides/hydroxides of iron and aluminum) and the phosphorus precipitation when combined with calcium and magnesium in alkaline conditions [4,5]. In tropical soil, oxisol is the soil type most commonly found and classified as weathered and acidic with high P sorption capacity and is intensively used for agriculture [6]. Withers et al. [7] showed that in acidic tropical soil, a rate of 35 kg P ha$^{-1}$ (or 80 kg ha$^{-1}$ P$_2$O$_5$) is required to overcome soil phosphate immobilization (wherein phosphate is adsorbed and/or precipitated). Oliveira et al. [8]

demonstrated that P usually presents low mobility in soil due to strong interaction with soil mineral constituents (e.g., hematite and gibbsite mineral). The low pH in acid soils improved the positive surface charge, which adsorbed the P-containing groups (e.g., inorganic $PO_4{}^{3-}$, $R–PO_4{}^{3-}$, phosphoryl, etc.) on the soil surface [9].

P fertilizers, commonly used to increase the soil P levels, are soluble, with a low use efficiency, and require extensive amounts of phosphate rocks and acids for their production [10]. According to International Fertilizer Associations Statistics [11], the global P fertilizer ($P_2O_5$) consumption for agriculture increased by 34.5 million tons from 1961 to 2019. In addition, crop P use efficiency is low, with an average between 9.1% and 12.4% [12]. Therefore, indicating that a large amount of P is applied and lost in agriculture. The P fertilizers associated with the organic matter are alternative fertilizers for increasing P efficiency in the soil with a significant increase in available P in tropical soils [13,14], improving the soil microbial biomass [15], root development, and yield of grains and legumes [16,17]. Chen et al. [18] showed that high levels of organic fertilizer improved soil P availability and crop growth of rice. Zhang et al. [19] showed combining manure (pig manure) with corn straw was more effective than applying these materials separately, promoting soil P transformation and availability. Alternative P sources with higher use efficiency are requested to build a circular economy, especially considering the recent evidence of excessive P use in tropical areas and high P demand for crop production [10].

Inoculation with Plant Growth-Promoting Bacteria (PGPB) also has been presented as another alternative to increasing the P availability in tropical soils leading to greater nutrient absorption by the plant, converting insoluble phosphorous forms to monobasic ($H_2PO_4{}^-$) and dibasic ($HPO_4{}^{2-}$) forms that are absorbable by plants [10]. Microorganisms present in PGPB can secrete acids (butyric, malic, acetic, lactic, citric acid, among others) or produce solubilize in organic soil phosphorous [20]. Estrada-Bonilla et al. [21] showed that inoculation with PGPB (including *Bacillus* sp. BACBR04, *Bacillus* sp. BACBR06, and *Rhizobium* sp. RIZBR01) increased the content of P in sugarcane shoots with an increment in soil P availability and changes in the soil bacterial community. Adnan et al. [22] showed that PGPB (including *Arthrobacter*, *Burkholderia*, *Bacillus*, *Enterobacter*, *Mycobacterium*, *Pseudomonas*, *Pantoea,* and *Rhizobia*) was a great alternative to increase the efficiency of mineral P fertilizers improving wheat yield in calcareous soil. Richardson and Simpson [23] demonstrated that there are several soil and rhizosphere bacteria displays that can promote plant growth and soil P levels.

Most bacterial genera of PGPBs include Acinetobacter, Agrobacterium, Arthobacter, Azospirillum, Azotobacter, Bradyrhizobium, Burkholderia, Frankia, Rhizobium, Serratia, Thiobacillus, Pseudomonas, and Bacillus [24]. The Pseudomonas and Bacillus are among the most extensively studied [25], and members of the genus Bacillus are popular inclusions in biostimulants [26] and biofertilizer formulations [27]. This popularity is attributable in part to the ability of Bacillus cells to form endospores, which are hardy and metabolically dormant cells capable of withstanding physical and chemical stressors (such as extreme heat, desiccation, ultraviolet radiation, and antimicrobial chemicals) that would kill other microbes [28].

PGPBs have a direct influence on root architecture and root growth which are important strategies of plants for exploration in soil for water and nutrients [29]. Specifically, for soil P acquisition by the plants, root growth is considered one of the most important factors [30,31]. Rosolem et al. [32] showed that the P applications increased soybean root growth under tropical conditions with a direct effect on grain production. These results are explained by the P influence on root architecture with modification in root meristematic cell activity [33]. Soybean and corn are widely cultivated, with a world production of 358.10 million tons (soybean) and 1214.88 million tons (corn) in the 2021/22 harvest [34] and with an average P demand of 8 kg of P ton$^{-1}$ of grain (soybean) and 4.6 kg of P ton$^{-1}$ of grain (corn) [35]. Brazil is considered a great producer of soybean and corn, with a respective area of 41,492.0 and 21,580.9 thousand hectares and a production of 125,549.8 and 113,133.6 thousand tons in 2021/22 harvest [36]. Therefore, based on the importance of

cereal production in food security and high P demand by the plants, strategies are requested to monitor the use of PGPB to increase P use efficiency, promote cereal production (soybean and corn), and combat soil nutrient depletion.

The goal of this study was to test the hypothesis that PGPB (*Bacillus subtilis*, *Bacillus amyloliquefaciens*, *Bacillus licheniformis*, and *Bacillus pumilus*) can be a strategy to increase the monoammonium phosphate efficiency, root growth, and P assimilation of soybean and corn cultivated in arenosol and oxisol.

## 2. Materials and Methods

### 2.1. Study Characterization

The study was conducted in a greenhouse located in the Escola Superior de Agricultura "Luiz de Queiroz" (ESALQ-USP), in Piracicaba, São Paulo, Brazil (22°42′ S, 47°38′ O), in 2020. The experimental design involved a randomized block design (RBD) with five rates of PGPB (0, 1.0, 1.33, 1.66, and 2.0 L per ton of fertilizer) sprayed on monoammonium phosphate (MAP; 11% of nitrogen, and 52% of phosphorus/$P_2O_5$) and applied in two soils (arenosol and oxisol) cultivated with soybean and corn.

In the study, there were five replications resulting in a total of 100 experimental units (5 PGPB rates ∗ 2 soils ∗ 2 crops ∗ 5 replications). A control group, treated as a control treatment, was used to monitor the application of monoammonium phosphate without PGPB. The PGPB rates (between 1 and 2 L per ton of fertilizer) were based on the quality of fertilizer, where rates higher than 2 L can increase the fertilizer moisture causing technical issues in the application.

The microbial solution of PGPB was sprayed on MAP fertilizer on the day of application. The microbial solution is a blend of proprietary microbial cultures (*Bacillus subtilis*: $2.5 \times 10^9$ cfu/mL; *Bacillus amyloliquefaciens*: $2.5 \times 10^9$ cfu/mL; *Bacillus licheniformis*: $2.5 \times 10^9$ cfu/mL; *Bacillus pumilus*: $2.5 \times 10^9$ cfu/mL), commercially known as BiOWiSH® Crop Liquid (BiOWiSH Technologies Inc., Cincinnati, OH, USA).

Soils were collected in the Piracicaba region, São Paulo, Brazil, with the soil taken from the top 0.2 m of the soil surface, at 4 locations within an area of 1 hectare. Soils were classified as an arenosol and an oxisol according to soil attributes in the Soil Taxonomy [37]. The soils were analyzed, and the following parameters were measured according to the soil methodologies described by Van Raij et al. [38].

Oxisol and arenosol presented a respective pH ($H_2O$) of 4.1 and 4.8 (classified as acid soils) with the content of the organic matter (Colorimetric Method) of 12.0 and 9.0 g dm$^{-3}$, phosphorus (extraction method, Mehlich 1) of 3.0 and 2.0 mg dm$^{-3}$, sulfur (US-EPA method) of 3.0 and 2.0 mg dm$^{-3}$, calcium (ion exchange resin method) of 9.0 and 11.0 mmolc dm$^{-3}$, magnesium (ion exchange resin method) of 4.0 and 6.0 mmolc dm$^{-3}$, and potassium (ion exchange resin method) of 0.3 and 0.5 mmolc dm$^{-3}$.

Soil texture also was measured using the pipette method, recommended by EMBRAPA [39], obtaining, as a result, a respective classification of sand texture (sand: 91.0%; silt: 0.9%; and clay: 8.1%) and loamy sand texture (sand: 81.0%; silt: 0.6%; and clay 18.4%) for arenosol and oxisol.

In the greenhouse, soils were sieved, dried, and corrected via the application of a dolomitic limestone filler (30% CaO, 20% MgO, and 100% of total neutralizing relative power) to increase base saturation to 60%. After 3 months, pots were filled with soil (4.0 kg soil pot$^{-1}$, the total volume of 5 dm$^3$) and fertilized with a nutrient solution containing nitrogen (150 mg kg$^{-1}$), sulfur (50 mg kg$^{-1}$), boron (1.5 mg kg$^{-1}$), manganese (3 mg kg$^{-1}$), zinc (5 mg kg$^{-1}$), molybdenum (0.1 mg kg$^{-1}$), and copper (1.5 mg kg$^{-1}$). Potassium was applied in the soil at a rate of 150 mg kg$^{-1}$ of potassium chloride.

In planting, PGPB + MAP was applied in soil using a rate of 200 mg kg$^{-1}$, which is considered a high rate of monoammonium phosphate application. Seeds of soybean (Cultivar DM66i68 IPRO) and corn (hybrid Seminis 1051) were planted (five seeds per pot) at a soil depth of 3 cm. Fifteen days after sowing, seedlings were thinned to two plants per pot, and another application of nutrient solution was applied. Throughout the experiment,

plants were irrigated every day with deionized water, and humidity was maintained at approximately 60% of the soil field capacity. The air temperature was maintained between 16 and 32 °C, and relative humidity was maintained between 44 and 81%.

### 2.2. Measurements

Samples of soil were collected for microbiological analyses on days 20 and 45 after the emergence of plants. In each sampling, 10 g of soil was collected on the surface at a soil depth of 0.2 m. The acid phosphatase and beta-glucosidase were determined according to Tabatabai [40]. The acid phosphatase (EC 3.1.3.2) was determined using solutions of buffer (at pH 6.5) and a soil-buffer mixture of p-nitrophenyl-phosphate (PNF). Meanwhile, the beta-glucosidase (EC 3.2.1.21) was determined using a solution of p-nitrophenyl-b-D-glucopyranoside (PNG) and p-nitrophenol. The ammonium-oxidizing microorganisms responsible for the ammonification process (nitrogen cycle stage) were quantified according to Woomer [41] using a dilution in a sodium chloride solution (NaCl; 1%). All enzymatic activities present close associations with the carbon, nitrogen, and phosphorus dynamics in the soil [42,43].

Plants were collected 45 days after their emergence and divided into the shoot (leaves and stem) and roots. Posteriorly, vegetal material was dried at a constant temperature of $65 \pm 2$ °C. The concentration of phosphorous was determined by nitroperchloric digestion followed by colorimetric assays [38]. The data of the P concentration and dry mass of the plant (above-ground and root parts) were used to calculate the P accumulation, multiplying the P concentration with dry mass in the corresponding part.

Soil samples were collected (10 g of soil per vase) on day 45 after the emergence of plants, and it was used to monitor the P content by the extraction method, Mehlich 1, according to Van Raij et al. [38]. Soil P contents were monitored on the last day based on previous studies [10–21] to monitor the soil P balance. The Analysis of Soil-Plant Analyses Development (SPAD) also was carried out using a chlorophyll meter, taking on 10 leaves per plot on day 45 after the emergence of plants.

### 2.3. Data Analysis

The Relative Efficiency Index (REI) was calculated using the data of dry mass production (above-ground and root parts) in the control (monoammonium phosphate without PGPB application) and for the treatments with applications of PGPB. The result was multiplied by 100 to find the percentage increment of dry matter production (Equation (1)).

$$\text{REI} \left( \text{g g}^{-1} \right) = \left( \text{DMP}_{\text{control}} \Big/ \text{DMP}_{\text{PGPB}} \right) \times 100 \tag{1}$$

where $\text{DMP}_{\text{control}}$ is the dry mass production in control without PGPB application (g), and $\text{DMP}_{\text{PGPB}}$ is the dry mass production in treatments with PGPB application (g).

Homogeneity of variance and normality of residuals were tested before carrying out the analysis of variance (ANOVA) using the Bartlett test and the Shapiro–Wilk test, respectively. There was no transformation to normalize data, according to the tests, which was expected due to control conditions for all experimental units in the greenhouse. In ANOVA, the rates of PGPB were submitted to analysis of variance (ANOVA) using the F-test ($p < 0.05$), and the averages were compared by the Duncan test when the F-test was significant ($p < 0.05$). General averages of soils (arenosol and oxisol) and crops (corn and soybean) were compared by the *t*-test (Student's *t*-test; $p < 0.05$) using the average between groups with paired samples. Variables were correlated by the Pearson test using a probability of 0.05. Statistical analyses were performed in R (version 4.0.0, Auckland, New Zeland; R Foundation for Statistical Computing), and the results were graphed in SigmaPlot (version 11.0; SYSTAT Software, Inc., Richmond, VA, USA).

## 3. Results

### 3.1. Biologic Activities in Soil

In corn cultivated in the arenosol, the highest activities of ammonium-oxidizing microorganisms and beta-glucosidase were recorded on day 20 at the rate of 2 L/ton of fertilizer (Figure 1).

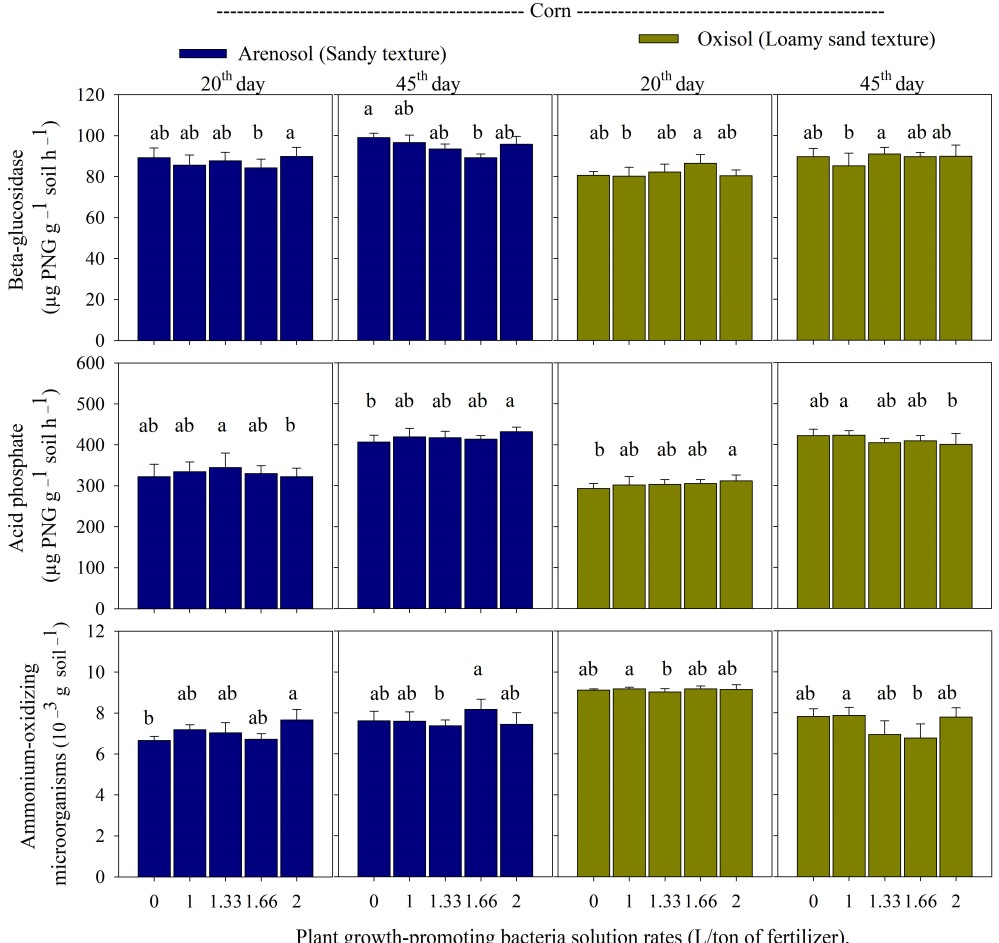

**Figure 1.** Beta-glucosidase, acid phosphatase, and ammonium-oxidizing microorganisms in an arenosol and oxisol cultivated with corn and application of monoammonium phosphate sprayed with rates of plant growth-promoting bacteria solution (PGPB; 0; 1; 1.33; 1.66; and 2.0 L/ton of fertilizer), in the periods of 20 and 45 days after corn emergence. Deviation bars indicate the standard error from the mean. Lowercase letters represent the comparison between treatments using the Duncan test ($p$-value < 0.05). Control is the application of monoammonium phosphate without PGPB.

There was a reduction of acid phosphatase on day 20 in arenosoil cultivated with corn. Meanwhile, on day 45, there was an inversion, with the higher activity of beta-glucosidase, and a reduction of acid phosphatase was observed in the control. In the oxisol with corn, the higher rate of PGPB + MAP promoted acid phosphatase on day 20, with a respective reduction on day 45 (Figure 1). The highest ammonium-oxidizing microorganisms were recorded at the rate of 1.0 L/ton of fertilizer on days 20 and 45 (Figure 1).

In the arenosol cultivated with soybean, the highest rate of PGPB + P increased the activity of beta-glucosidase and ammonium-oxidizing microorganisms on days 20 and 45 (Figure 2).

For the acid phosphatase, the highest rate of PGPB + MAP presented a higher acid phosphatase on day 20, while higher acid phosphatase was recorded in the control on day 45 in the arenosol cultivated with soybean (Figure 2). In the oxisol, the highest rate of PGPB + MAP promoted beta-glucosidase on day 20, while low levels of acid phosphatase

activity were observed. Furthermore, increased ammonium-oxidizing microorganisms were observed with the highest rate on day 45 (Figure 2).

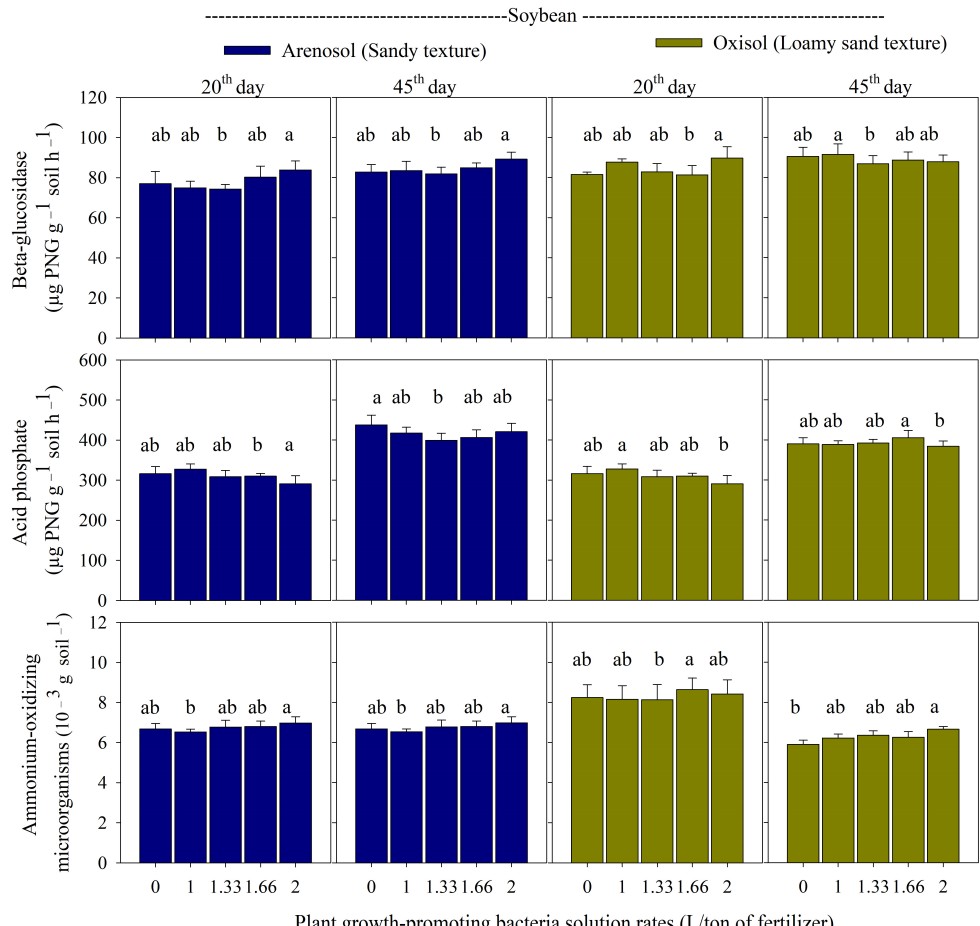

**Figure 2.** Beta-glucosidase, acid phosphatase, and ammonium-oxidizing microorganisms in an arenosol and oxisol cultivated with soybean and application of monoammonium phosphate sprayed with rates of plant growth-promoting bacteria solution (PGPB; 0; 1; 1.33; 1.66; and 2.0 L/ton of fertilizer), in the periods of 20 and 45 days after soybean emergence. Deviation bars indicate the standard error from the mean. Lowercase letters represent the comparison between treatments using the Duncan test ($p$-value < 0.05). Control is the application of monoammonium phosphate without PGPB.

### 3.2. Plant Measurements

Corn height was directly influenced by the increased rates of PGPB + MAP when cultivated in the arenosol at the rates between 1.66 and 2.00 L/ton of fertilizer, while there was no effect on corn cultivated in the oxisol (Table 1). In the soil characterization, oxisol and arenosol presented a respective pH ($H_2O$) of 4.1 and 4.8, classified both as acid soils. In both soils, plant height presented a positive correlation with the dry matter of the shoot (r = 0.94; $p$ < 0.05) and roots (r = 0.87; $p$ < 0.05).

Interestingly, the PGPB + MAP at the rate of 1.66 L/ton of fertilizer influenced the SPAD of corn in the oxisol, while there was no effect observed in the arenosol. For both SPAD and corn height, the control presented lower values with a height of 161.5 cm in the arenosol and a SPAD of 41.5 in the oxisol (Table 1).

In both soils, the total dry matter of corn increased in parallel to the rates of PGPB + MAP, representing an increase of 10% and 26%, respectively, in the arenosol and the oxisol, when compared to the control at the rate of 2.0 L/ton of fertilizer (Table 1). The highest rate of PGPB + MAP also promoted the REI of the above-ground part, with an increase of 30% compared with the control (Table 1).

**Table 1.** Height, SPAD, dry mass (shoot and root), relative efficiency index (REI) of corn cultivated in an arenosol and oxisol with the application of monoammonium phosphate sprayed with rates of plant growth-promoting bacteria (PGPB; 0; 1; 1.33; 1.66; and 2.0 L/ton of fertilizer).

| PGPB | Corn | | | | | | |
| --- | --- | --- | --- | --- | --- | --- | --- |
| Rates | Height | SPAD | Dry Mass (g Vase$^{-1}$) | | | REI (%) | |
| L/t | cm | | Shoot | Roots | Total | Shoot | Roots |
| | | | | Arenosol | | | |
| 0.00 | 161.5 ± 4.1 B | 40.6 ± 1.8 A | 44.3 ± 2.0 A | 17.4 ± 0.7 B | 61.8 ± 1.7 B | - | - |
| 1.00 | 168.6 ± 4.0 AB | 39.8 ± 0.9 A | 42.4 ± 1.3 A | 13.9 ± 0.2 C | 56.3 ± 1.3 C | 101.3 ± 2.6 A | 73.1 ± 15.2 A |
| 1.33 | 175.5 ± 3.7 A | 43.4 ± 1.6 A | 45.5 ± 1.7 A | 13.2 ± 0.5 C | 58.8 ± 1.6 BC | 105.6 ± 4.8 A | 72.8 ± 13.7 A |
| 1.66 | 173.8 ± 4.0 A | 42.8 ± 0.7 A | 45.4 ± 1.9 A | 16.3 ± 0.8 B | 61.8 ± 1.7 B | 107.4 ± 4.8 A | 84.5 ± 12.0 A |
| 2.00 | 169.2 A ± 1.8 B | 42.4 ± 0.5 A | 41.5 ± 1.0 A | 26.5 ± 1.3 A | 68.1 ± 1.6 A | 99.1 ± 2.0 A | 104.0 ± 12.3 A |
| CV% | 3.7 | 6.4 | 7.1 | 9.6 | 5.0 | 13.4 | 44.5 |
| | | | | Oxisol | | | |
| 0.00 | 157.1 ± 8.3 A | 41.5 ± 1.7 B | 43.2 ± 2.0 B | 14.0 ± 0.5 C | 57.3 ± 2.1 C | - | - |
| 1.00 | 142.3 ± 5.8 A | 43.3 ± 1.3 AB | 36.8 ± 1.4 C | 17.7 ± 0.7 B | 54.6 ± 1.8 C | 88.4 ± 3.2 B | 111.6 ± 13.3 A |
| 1.33 | 159.3 ± 5.9 A | 42.2 ± 0.8 AB | 43.6 ± 1.7 B | 19.8 ± 0.9 B | 63.5 ± 2.1 B | 100.8 ± 3.9 AB | 118.8 ± 14.6 A |
| 1.66 | 152.9 ± 6.0 A | 45.5 ± 1.2 A | 40.1 ± 1.7 BC | 19.5 ± 0.5 B | 59.7 ± 1.4 BC | 92.8 ± 3.9 B | 102.0 ± 14.5 A |
| 2.00 | 154.1 ± 8.2 A | 43.9 ± 1.5 AB | 51.2 ± 2.4 A | 26.4 ± 1.4 A | 77.7 ± 3.0 A | 125.3 ± 1.2 A | 153.9 ± 14.0 A |
| CV% | 8.7 | 6.1 | 8.0 | 9.3 | 6.6 | 21.8 | 36.5 |

CV: coefficient of variation; averages with the standard error; uppercase letters represent the comparison between treatments (PGPB rates) using the Duncan test ($p$-value < 0.05). The general averages of soils and crops were tested by the $t$-test ($p$-value < 0.05). Control is the application of monoammonium phosphate without PGPB.

In the arenosol, soybean was not affected by the application of PGPB + MAP on dry matter; however, in the oxisol, it was evident that the application of PGPB + MAP increased the dry matter of roots (up to 31%) and total (up to 8%) as compared to the control. The REI also was higher in the group with the highest rate of 2.0 L/ton of fertilizer in soybean cultivated in the oxisol (Table 2).

**Table 2.** Height, SPAD, dry mass (shoot and root), relative efficiency index (REI) of corn cultivated in an arenosol and oxisol with the application of monoammonium phosphate sprayed with rates of plant growth-promoting bacteria (PGPB; 0; 1; 1.33; 1.66; and 2.0 L/ton of fertilizer).

| PGPB | Soybean | | | | | | |
| --- | --- | --- | --- | --- | --- | --- | --- |
| Rates | Height | SPAD | Dry Mass (g Vase$^{-1}$) | | | REI (%) | |
| L/t | cm | | Shoot | Roots | Total | Shoot | Roots |
| | | | | Arenosol | | | |
| 0.00 | 77.2 ± 4.1 A | 33.0 ± 1.2 A | 16.2 ± 0.8 A | 1.9 ± 0.1 C | 18.1 ± 0.4 A | - | - |
| 1.00 | 74.7 ± 4.0 A | 32.6 ± 1.0 A | 16.1 ± 0.6 A | 2.5 ± 0.1 B | 18.7 ± 0.3 A | 88.9 ± 4.4 A | 92.4 ± 9.1 A |
| 1.33 | 74.6 ± 3.7 A | 33.7 ± 0.5 A | 15.7 ± 0.6 A | 3.0 ± 0.1 A | 18.8 ± 0.3 A | 94.9 ± 4.0 A | 109.6 ± 16.3 A |
| 1.66 | 77.0 ± 4.0 A | 33.3 ± 0.6 A | 15.6 ± 0.5 A | 2.5 ± 0.1 B | 18.2 ± 0.3 A | 81.7 ± 4.1 A | 91.3 ± 12.6 A |
| 2.00 | 69.1 ± 1.8 A | 33.6 ± 0.7 A | 16.6 ± 0.5 A | 2.5 ± 0.1 B | 19.2 ± 0.3 A | 89.1 ± 4.3 A | 91.5 ± 8.4 A |
| CV% | 8.2 | 4.8 | 7.2 | 7.1 | 6.5 | 16.4 | 23.4 |
| | | | | Oxisol | | | |
| 0.00 | 54.8 ± 2.1 AB | 32.8 ± 1.0 A | 18.3 ± 0.9 A | 2.4 ± 0.1 D | 20.7 ± 1.0 B | - | - |
| 1.00 | 50.2 ± 2.0 B | 31.3 ± 0.8 A | 18.5 ± 0.8 A | 3.3 ± 0.1 AB | 21.9 ± 0.9 AB | 94.9 ± 2.6 AB | 120.2 ± 22.2 AB |
| 1.33 | 51.1 ± 1.1 B | 31.1 ± 0.9 A | 16.0 ± 0.5 B | 2.8 ± 0.1 C | 19.0 ± 0.6 C | 87.8 ± 4.7 B | 110.6 ± 13.7 B |
| 1.66 | 56.5 ± 2.5 A | 32.3 ± 1.7 A | 17.5 ± 1.0 A | 3.1 ± 0.1 BC | 20.6 ± 1.1 BC | 92.8 ± A4.8 B | 115.8 ± 11.9 AB |
| 2.00 | 56.0 ± 1.8 A | 34.0 ± 0.5 A | 18.8 ± 0.8 A | 3.5 ± 0.1 A | 22.4 ± 0.9 A | 99.6 ± 2.0 A | 136.4 ± 24.6 A |
| CV% | 6.6 | 6.9 | 5.9 | 7.3 | 5.7 | 8.6 | 14.6 |
| | General Average | | | | | | |
| | Height | | Dry mass (g vase$^{-1}$) | | | REI (%) | |
| | cm | | Shoot | Roots | Total | Shoot | Roots |
| Soils | | | | | | | |
| Arenosol | 122.1 ± 6.8 A | | 29.9 ± 2.0 A | 10.0 ± 8.3 A | 39.9 ± 3.0 A | 95.7 ± 1.1 A | 89.7 ± 5.5 A |
| Oxisol | 103.4 ± 7.2 A | | 30.4 ± 1.9 A | 11.2 ± 1.2 A | 41.7 ± 3.1 A | 121.2 ± 1.2 A | 91.5 ± 5.7 A |
| Crops | | | | | | | |
| Corn | 161.4 ± 1.9 A | | 43.4 ± 0.6 A | 18.5 ± 0.6 A | 61.9 ± 1.0 A | 92.1 ± 1.3 A | 108.4 ± 6.6 A |
| Soybean | 64.1 ± 1.7 B | | 16.9 ± 0.2 B | 2.7 ± 0.0 B | 19.7 ± 0.2 B | 89.0 ± 1.2 A | 108.5 ± 4.0 A |

CV: coefficient of variation; averages with the standard error; uppercase letters represent the comparison between treatments (PGPB rates) using the Duncan test ($p$-value < 0.05). The general averages of soils and crops were tested by the $t$-test ($p$-value < 0.05). Control is the application of monoammonium phosphate without PGPB.

In general, root dry matter was increased by the application of PGPB + MAP in both soils and plants when compared with above-ground development. Based upon the general average, the PGPB + MAP increased the dry matter of roots in corn cultivated in the oxisol and soybean cultivated in both soils compared to the control (Figure 3).

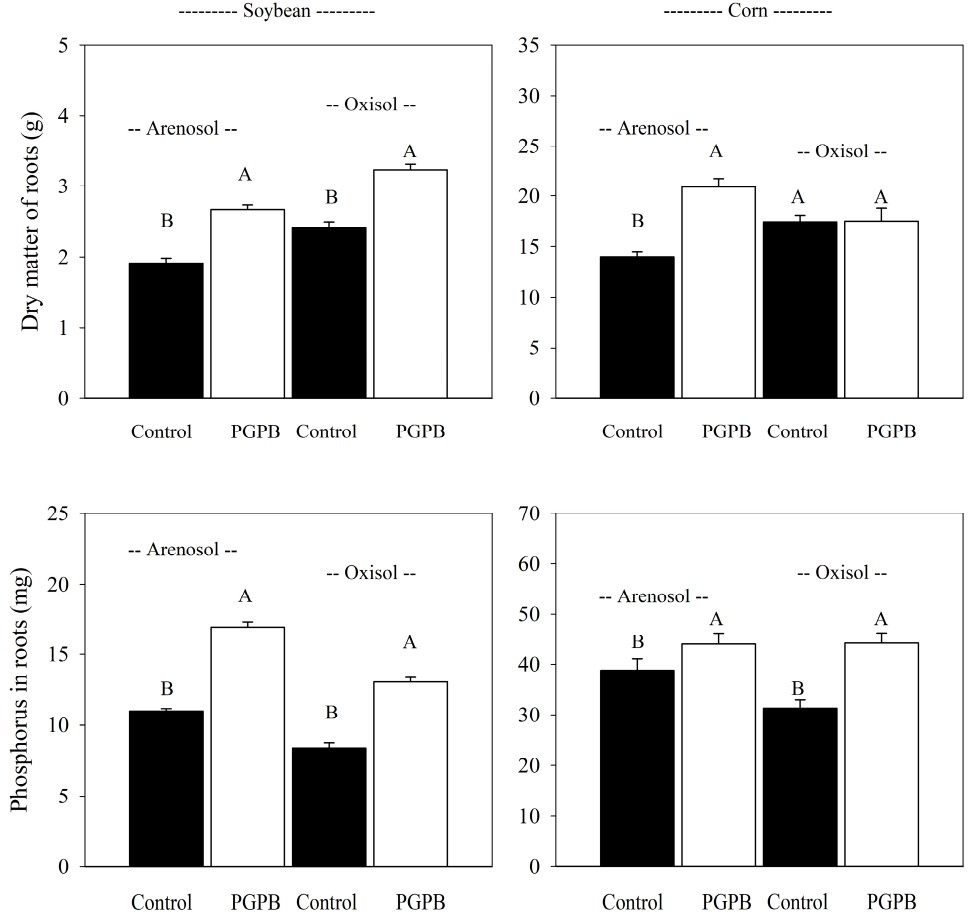

**Figure 3.** General average of root dry matter and phosphorus in roots of corn and soybean cultivated in an oxisol and arenosol with the application of monoammonium phosphate sprayed with plant growth-promoting bacteria (PGPB). Deviation bars indicate the standard error from the mean. The general averages of soils and crops were tested by the *t*-test (*p*-value < 0.05), and different uppercase letters represent the comparison between averages.

There were positive correlations between the dry matter of roots and above-ground material (r = 0.90; $p < 0.05$), indicating a positive root-developing influence on above-ground development.

### 3.3. P in Soils and Plants

In the arenosol, PGPB + MAP decreased the phosphorous content in soil cultivated with soybean and corn, indicating that the phosphorous applied to the soil was absorbed by plants. Consequently, the higher phosphorus accumulation in shoot and roots were monitored in plants with the application of PGPB + MAP (Table 3).

Also, there was a reduction in phosphorous in the soil as a consequence of phosphorous accumulation in plants in the oxisol cultivated with corn. However, this result was not observed in soil cultivated with soybean with higher accumulation of phosphorous in the control and PGPB + MAP (Table 3).

Additionally, the accumulation of phosphorous in roots exhibited a positive influence on the dry matter of roots with an r of 0.96 ($p < 0.05$), which had a higher correlation than the influence of phosphorous accumulation in above-ground dry matter with an r of 0.55 ($p < 0.05$).

**Table 3.** Phosphorus (P) in soil (arenosol and oxisol) and plants (shoot and root parts) with the application of monoammonium phosphate sprayed with rates of plant growth-promoting bacteria (PGPB; 0; 1; 1.33; 1.66; and 2.0 L/ton of fertilizer), cultivated with corn and soybean.

| | Corn | | | Soybean | | |
|---|---|---|---|---|---|---|
| **PGPB** | **P in Soil** | **P in Shoot** | **P in Roots** | **P in Soil** | **P in Shoot** | **P in Roots** |
| **Rates** | **mg dm$^{-3}$** | **mg Vaso$^{-1}$** | | **mg dm$^{-3}$** | **mg Vaso$^{-1}$** | |
| L/t | | | | Arenosol | | |
| 0.00 | 135.6 ± 1.4 A | 163.9 ± 4.9 C | 38.8 ± 2.3 C | 120.8 ± 4.2 A | 99.7 ± 4.0 C | 11.0 ± 0.1 D |
| 1.00 | 110.8 ± 4.5 B | 172.4 ± 1.7 BC | 39.6 ± 2.6 C | 98.4 ± 5.5 B | 111.7 ± 3.1 AB | 15.3 ± 0.6 C |
| 1.33 | 112.6 ± 2.7 B | 182.5 ± 4.7 B | 32.7 ± 1.1 D | 84.6 ± 5.2 C | 106.4 ± 2.3 BC | 17.1 ± 0.5 B |
| 1.66 | 93.8 ± 4.8 D | 216.8 ± 9.7 A | 46.5 ± 1.6 B | 82.4 ± 4.9 C | 116.9 ± 3.5 A | 19.0 ± 0.3 A |
| 2.00 | 101.6 ± 3.2 CD | 160.6 ± 5.6 C | 57.1 ± 1.2 A | 120.6 ± 4.8 A | 112.8 ± 3.4 AB | 16.3 ± B0.3 C |
| CV% | 6.30 | 7.48 | 9.38 | 9.31 | 6.80 | 6.26 |
| | | | | Oxisol | | |
| 0.00 | 135.0 ± 0.8 A | 69.1 ± 3.6 C | 31.3 ± 1.6 D | 68.2 ± 4.9 B | 62.9 ± 3.2 A | 8.4 ± 0.3 C |
| 1.00 | 126.8 ± 2.9 B | 83.3 ± 4.3 B | 37.2 ± 1.4 C | 84.2 ± 2.1 A | 66.8 ± 2.8 A | 13.9 ± 0.5 A |
| 1.33 | 101.8 ± 2.6 C | 113.0 ± 2.7 A | 37.2 ± 1.1 C | 63.8 ± 4.1 B | 52.7 ± 2.0 B | 11.4 ± 0.4 B |
| 1.66 | 129.6 ± 3.1 AB | 89.3 ± 4.1 B | 44.9 ± 1.6 B | 84.8 ± 4.3 A | 65.1 ± 3.3 A | 13.2 ± 0.4 A |
| 2.00 | 131.8 ± 2.1 AB | 109.7 ± 2.8 A | 57.4 ± 1.8 A | 69.0 ± 5.5 B | 70.3 ± 2.8 A | 13.6 ± 0.3 A |
| CV% | 4.22 | 6.74 | 9.02 | 8.57 | 10.64 | 8.77 |
| | | | General Average | | | |
| | P in soil | P in shoot | P in roots | P in soil | P in shoot | P in roots |
| | mg dm$^{-3}$ | mg vaso$^{-1}$ | | | mg dm$^{-3}$ | |
| Soils | | | | Crops | | |
| Arenosol | 106.1 ± 2.5 A | 144.3 ± 5.5 A | 29.3 ± 2.1 A | Corn | 117.9 ± 2.4 A | 136.0 ± 3.2 A | 42.2 ± 0.3 A |
| Oxisol | 99.5 ± 4.4 B | 78.2 ± 4.4 B | 26.8 ± 2.3 A | Soybean | 87.6 ± 2.9 B | 86.5 ± 3.5 B | 13.9 ± 0.4 B |

CV: coefficient of variation; averages with the standard error; uppercase letters represent the comparison between treatments (PGPB rates) using the Duncan test ($p$-value < 0.05). The general averages of soils and crops were tested by the $t$-test ($p$-value < 0.05). Control is the application of monoammonium phosphate without PGPB.

## 4. Discussion

In both soils and crops, there was a variation in soil biological activity during this experiment, which was expected and well-understood based on enzymatic activity resulting in the rapid modification of soil, as demonstrated by Aşkın et al. [44], Veeraragavan et al. [45], and Margalef et al. [46]. The addition of fertilizers readily alters the soil's enzymatic activities due to the high availability of nutrients which is well-known [47,48]. As observed on day 45, PGPB + MAP promoted beta-glucosidase, ammonium-oxidizing microorganisms, and phosphatase activities. These results are important because the soil enzymatic activities are considered great indicators of soil quality [49]. Soil enzymes can be used as soil quality indicators due to their quick response to changes in environmental stress, pollution, and agricultural practices. Considered a better soil quality indicator than organic matter and soil physical and chemical properties [50]. The beta-glucosidase activity is related to the carbon cycle acting in the cleavage of cellobiose into glucose molecules [42–51]. Meanwhile, ammonium-oxidizing microorganisms are responsible for the nitrification process in the conversion of $NH_4^+$-N into hydroxylamine and further to $NO_3^-$-N [52,53], and phosphatase is related to the transformation of complex and unavailable forms of organic P into assimilable phosphate [54]. These extracellular enzymes are produced by bacteria, fungi, and/or plant roots and serve to cleave the nutrient group from its substrates [55–57].

The increased enzymatic activities observed in this experiment on day 45 over day 65 indicate that PGPB + P promoted the r-strategy groups in the soil microbiota, characterized by a fast growth rate caused by the increase of microorganisms in the soil [58]. Lopes et al. [10] also showed that in soil with the addition of compost enriched by phosphorite (P easily released) and PGPB, the r-strategy groups prevailed in the soil microbiota due to the easy degradation of the substrate. However, in compost enriched by apatite, the k-strategy (slow growth rates) prevailed due to the low release of P in apatite. In the study, r-strategy groups were predominant due to the high availability of nutrients from fertilizers with fast realizing to the soil. In addition, many *Bacillus* species are strongly plant-associated as endophytes, known to mediate a variety of positive outcomes in their host plants through the production of phytohormones and volatile organic compounds with the

mediation of critical plant processes such as water transport, pigment, and hormone synthesis, and the expression of stress-response genes [59]. Multiple studies have demonstrated the effects of *Bacillus* in agriculture, which increase nutrient solubilization and seedling emergence rate [60,61], as well as inhibit the growth of soil and leaf pathogens [62,63]. However, the practical use of PGPBs in agriculture using formulated products currently represents a small fraction of agricultural practices. The low survival of organisms in the field is presented as a challenge in the application of PGPBs in agriculture [64]. It is also important to mention that the selection of growth-promoting bacteria is critical because the response of the plant varies depending on the bacterial isolate, the genotype of the plant, and the conditions of the experiment. In the study, a solution with *Bacillus subtilis* ($2.5 \times 10^9$ cfu/mL), *Bacillus amyloliquefaciens* ($2.5 \times 10^9$ cfu/mL), *Bacillus licheniformis* ($2.5 \times 10^9$ cfu/mL), and *Bacillus pumilus* ($2.5 \times 10^9$ cfu/mL), commercially known as BiOWiSH® Crop Liquid (BiOWiSH Technologies Inc., Cincinnati, OH, USA), was tested which was mixed on monoammonium phosphate a common industrial fertilizer with two constituents (nitrogen and phosphorus) and rapid P-release.

The acid phosphatase activity was lower in some measurements at the rate of 2.0 L/ton of fertilizer, demonstrating that the PGPB + MAP did not stimulate the mineralization of organic phosphorous. Based on these measurements, and decrease in acid phosphatase activity was observed, which may be attributed to the high solubility of monoammonium phosphate (solubility at 20 °C; 370 g/L water). Probably, the efficiency of PGPB may be higher when associated with insoluble P sources, as demonstrated in other studies [65]. Lopes et al. [10] showed a positive and clear effect of phosphate-solubilizing bacteria inoculation when associated with phosphate rocks of low solubility (apatite), where the major part of phosphorous was bound to calcium. Lin et al. [62] demonstrated that poultry litter with *Bacillus* spp. increased corn yield and biomass compared to mineral fertilizer. Meanwhile, Billah and Bano [63] showed that PGPB also was applied to wheat seeds promoting height, P uptake, and yield of grain.

In arenosol, the PGPB + MAP was more responsive to biological activities in this experiment, suggesting that the PGPB presents a higher efficiency in light soils. Aşkın et al. [44] showed that the soil enzymes were increased in coarse-textured soils but not observed in soils of high clay or organic matter content. Soil particle size distribution, especially clay content, is a major factor affecting soil enzymatic activities. The influence of clay can be attributed to the adsorption capacity of clay, wherein extracellular enzymes are adsorbed by the clay's surface area [64]. Margalef et al. [46] showed that phosphatase and urease were reduced in clay soil due to the adsorption of the enzyme on the clay's surface area. However, Tietjen et al. [64] noticed that the beta-glucosidase activity was increased in soil with the adsorption to montmorillonite and organic matter contents. The beta-glucosidase activity is associated with organic matter content (that also presents adsorbed surface area) with a positive relationship and explains this outcome.

The PGPB + MAP rates promoted the corn height in the arenosol (which presents lower acid), while there was no influence in oxisol. For soybean, the PGPB + MAP rates promoted the height in the oxisol, while there was no influence in arenosol. In the soil characterization, oxisol and arenosol were classified as acid soils with respective pH (H$_2$O) of 4.1 and 4.8, characterized by phosphorous adsorption. The pH condition is considered the most suitable setting for improving acid phosphatase and organic acids production. Abdelgalil et al. [65] demonstrated that maximal acid phosphatase production was noticed with a pH of 7.5 rather than a pH of 1.0 N HCl. Musarrat et al. [66] demonstrated that neutral or slightly acidic pH was optimal for bacteria to solubilize organic phosphate. Bueis et al. [67] showed that the soil pH and water availability seem to be major constraints for enzyme activities. In the study, the water effect was not monitored in enzyme activities because the plants were irrigated every day with deionized water, and humidity was maintained at approximately 60% of the soil field capacity at both soils. Therefore, plant height was associated with better soil conditions caused by the PGPB application. The rate of 1.0 L/ton of microbial solution recorded the highest biological activity, considered an

optimal alternative without negative impact on fertilizer moisture that can cause technical issues in the application.

The application of PGPB + MAP also promoted the soil P contents from 3.0 to 99.5 mg dm$^{-3}$ (Oxisol) and from 2.0 to 106.1 mg dm$^{-3}$ (arenosol), clearly demonstrating a content higher than 100% on soil P available from the soil characterization (before the experiment). Applications of dolomitic limestone filler (30% CaO; 20% MgO; and 100% PRNT) also were requested to increase base saturation to 60% as well as to increase the pH, which also impacted soil P. In both soils, the total dry matter was increased by the rates of PGPB + MAP, which demonstrate a clear P accumulation in roots. The accumulation of phosphorous in roots was higher than the phosphorous accumulation in above-ground dry matter. These results indicate that PGPB + MAP improves root growth and builds strong root systems for nutrient and water exploration. A high root:shoot biomass ratio is a characteristic of roots that facilitates soil exploration and nutrient uptake [68]. Previous studies also demonstrated similar results in stimulating root growth with P and PGPB applications [69–71]. El Zemrany et al. [72] showed that PGPB Azospirillum lipoferum sprayed on corn seeds increased root development and root system architecture with a positive effect on corn development. Arkhipova et al. [73] showed that bacterization of the seeds of (bacteria *Bacillus subtilis* IB-21 and B. subtilis IB-22 and gram-negative bacteria Advenella kashmirensis IB-K1 and Pseudomonas extremaustralis IB-K13-1A) increased phosphate mobility in the rhizosphere and auxin content with a positive impact on wheat crop yield. Richardson et al. [74] explain that the interaction of roots with soil microorganisms is associated with nutrient availability and through the mechanisms that are associated with plant growth promotion. Higher root development is a very desirable condition in tropical agriculture because greater rooting promotes greater tolerance to drought and also better utilization of nutrients and water from the soil.

## 5. Conclusions

The inoculation with Plant Growth-Promoting Bacteria (compost *Bacillus subtilis*, *Bacillus amyloliquefaciens*, *Bacillus licheniformis*, and *Bacillus pumilus*) sprayed on monoammonium phosphate is a great alternative to increase the root growth of soybean and corn cultivated in arenosol and oxisol. In both soils and crops, there were varied results on the activities of beta-glucosidase, acid phosphatase, and ammonium-oxidizing microorganisms during the experiment, caused by the increase in rates of growth-promoting bacteria and indicating a better soil quality. There was a direct reduction in soil phosphorous content and demonstrated an increase in plant phosphorus accumulation. The highlights of the study were that (i) the rate of 1.0 L/ton of microbial solution recorded the highest biological activity without technical issues in the application; (ii) the microbial solution improved the soil biological activity and plant phosphorus accumulation; (iii) the crop root growth was increased by microbial solution in both soil types with direct on plant development. Based on these results, the authors conclude that the monoammonium phosphate sprayed growth-promoting bacteria increases the growth and P accumulation of soybean and corn cultivated in arenosol and oxisol soils, with a direct effect on crop rooting in greenhouse conditions. Future studies could monitor the indigenous phosphorus-solubilizing microorganisms in the soil to elucidate the association with the microbial solution of growth-promoting bacteria.

**Author Contributions:** C.P.S., F.D.A., J.C., J.G. and R.O. Data curation: C.P.S., F.D.A., R.F.-A. and R.O. Investigation and Methodology: C.P.S., F.D.A. and R.O. Writing—original draft and Writing—review & editing: R.F.-A., C.P.S., F.D.A., J.C., J.G. and R.O. All authors have read and agreed to the published version of the manuscript.

**Funding:** The APC was funded by the BiOWiSH Technologies.

**Institutional Review Board Statement:** Not applicable.

**Informed Consent Statement:** Not applicable.

**Data Availability Statement:** Not applicable.

**Acknowledgments:** Thanks to the Coordenação de Aperfeiçoamento de Pessoal de Nível Superior (CAPES; grant number 88882.317567/2019-01), the University of São Paulo "Luiz de Queiroz" College of Agriculture, and BiOWiSH Technologies for technical and financial support. We also thank Michelle Langefeld for all comments during the project.

**Conflicts of Interest:** The authors declare no conflict of interest.

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
