# Peer review of "Microbial Solution of Growth-Promoting Bacteria Sprayed on Monoammonium Phosphate for Soybean and Corn Production"

_agronomy, doi:10.3390/agronomy13020581_

Round 1

Reviewer 1 Report

Dear Corresponding Author

Hope you are doing well. I checked your paper and I have some comments:

1) In biological works, normality of data is not common and in Kurtosis and Skewness coefficient show that normal distribution of data is not OK. Anyway, you simply can analyze your data with non-parametric analysis such as Kruskal-Wallis.

2) In Figure 3 you used Duncan test to compare 2 groups in Oxisol and in Arenosol. It is impossible. Duncan test are used in at least 3 groups. For 2 groups you should do T-test.

3) In Figure 3, in dry matter of roots 'Arenosol' there is no letter (a, b, ab, ...?).

4) In all of figures you need to add standard error NOT standard deviation.

5) In all of tables you need to add average±standard error.

Generally I think you will need to re-analyze statistics of the paper.

Regards

Author Response

Reviewer: In biological works, normality of data is not common and in Kurtosis and Skewness coefficient show that normal distribution of data is not OK. Anyway, you simply can analyze your data with non-parametric analysis such as Kruskal-Wallis. In Figure 3 you used Duncan test to compare 2 groups in Oxisol and in Arenosol. It is impossible. Duncan test are used in at least 3 groups. For 2 groups you should do T-test.

Authors: Our study was developed in greenhouse conditions with control conditions, explaining the normal distribution of data. The homogeneity of variance and normality of residuals were tested before carrying out the analysis of variance (ANOVA), using the Bartlett test and the Shapiro-Wilk test, respectively. There was no transformation to normalize data, according to the tests, which was expected due to control conditions for all experimental units in the greenhouse. In ANOVA, the rates of PGPB were submitted to analysis of variance (ANOVA) using the F-test (p < 0.05), and the averages were compared by the Duncan test when the F-test was significant (p < 0.05). While general averages of soils (arenosol and) and crops (corn and soybean) were compared by the t-test (Student's t-test; p < 0.05) using the average between groups with paired samples. We added that information in the Material and Methods and/or Figures to leave a clear message to our readers. Thank you for the comments.

Reviewer: In Figure 3, in dry matter of roots 'Arenosol' there is no letter (a, b, ab, ...?). In all of the figures, you need to add standard error NOT standard deviation. In all of tables you need to add average±standard error.

Authors: We added the letters in Figure 3, there was no difference in the dry matter of roots 'Arenosol'. In the Tables and Figures, we changed the standard deviation to standard error in all Figures and tables. Also, the Figures were edited based on your comments and other reviewers. Thanks for taking the time to read our manuscript.

Reviewer 2 Report

This manuscript is very long, please summarized and shorten it.
There are some grammatical mistakes in the text. Please check and correct them.
The methodology is designed well.
The results and Discussion sections are enough.
Some of the literature is very old, please change them with new ones (after 2000 years).
Please check the pdf file and correct
some mistakes.

The plagiarism rate has been determined % 17, which is enough

Author Response

Reviewer: “This manuscript is very long, please summarized and shorten it. There are some grammatical mistakes in the text. Please check and correct them. The methodology is designed well. The results and Discussion sections are enough.
Some of the literature is very old, please change them with new ones (after 2000 years). please check the pdf file and correct some mistakes. The plagiarism rate has been determined % 17, which is enough

Authors: Thank you for all your comments. The manuscript was edited based on your comments and the other reviewer. We added more information in the Introduction based on the suggestion of other reviewers. Grammatical mistakes in the text were checked and citations 35 and 36 were removed. However, we would like to keep references 31 and 32 because there are classic and methodologic descriptions for microbiological analyses (Tabatabai, 1994; Woomer, 1994), and there was no other with the same description. We hope you understand our point of view. The reference list were updated. Thank you for your recommendations.

Reviewer: “The expression g kg-1 for the texture expression of soils is not very common. Please indicate the sand, silt, and clay content of the soils in % (Example: 91% Sand, 0.9% silt and 8.1% clay).”; “The expression aero parts for above the ground of plants is not very common. Please use shoot and root mass instead of aero parts.”

Authors: The editions were made and texture expression was edited to %. We also changed the expression “above” to shoot in the text. Thanks

Reviewer: This reference is old date; please change it with new one (31-32 and 35-36)

Authors: The references 35 and 36 were removed. However, we would like to keep references 31 and 32 because there are classic and methodologic descriptions for microbiological analyses (Tabatabai, 1994; Woomer, 1994), and there was no other with the same description. We hope you understand our point of view. The references were updated. Thank you for your recommendations.

Reviewer 3 Report

Review report for Agronomy 

General comments: 

This study is very interesting and has a scientific topic with a great impact on the field. The manuscript will be suitable for publication after taking care of the following minor comments.

Detailed comments:

1-The English language and /writing style is fine needs some minor check spelling and grammar check

2-Please avoid using the personal pronouns (I, We,) such as in line102 (in our study )and more 

Abstract

_This section is well written and the aim of the study was directly stated. 

Keywords:

-The keywords has been chosen very carefully and accurately but 

- please add soybean and corn to the keywords list

Introduction

-The introduction doesn’t provide sufficient background and it is missing enough relevant references

-This section needs to be elongated and enriched with more background about this topic.

Materials and Methods

-it is ok and adequate 

 Results:

The results are very interesting BUT the data presentation needs more attention especially Figure 1,2,and3) the resolution is very low the author is advised to provide NEW figures with better resolution.

Discussion:

*This section is poorly written.

the author is strongly advised to combine the results and discussion in one section for better interpretation and discussion for the presented data.

**Please rewrite and discuss in details, and fully discussed with related citations.

Conclusion 

.This section is well written and the conclusion is supported by the results of this study and please includes more data from the results, the most important findings. This section needs to include more application in different ways with a wider range for more crops in general.

References

This section is well written and it is up To date. 

Author Response

Reviewer, General comments: This study is very interesting and has a scientific topic with a great impact on the field. The manuscript will be suitable for publication after taking care of the following minor comments.

Authors: Thank you for all your comments. The manuscript was edited based on your comments.

Reviewer, Detailed comments:1-The English language and /writing style is fine needs some minor check spelling and grammar check; 2-Please avoid using the personal pronouns (I, We,) such as in line102 (in our study ) and more. Abstract _This section is well written and the aim of the study was directly stated. Keywords: -The keywords has been chosen very carefully and accurately but - please add soybean and corn to the keywords list

Authors: We deleted the personal pronouns (I, We,) in the text and added the soybean and corn (scientific name) to the keywords list.

Reviewer, Introduction: -The introduction doesn’t provide sufficient background and it is missing enough relevant references. -This section needs to be elongated and enriched with more background about this topic. Materials and Methods -it is ok and adequate Discussion: *This section is poorly written the author is strongly advised to combine the results and discussion in one section for better interpretation and discussion of the presented data.

Authors: We enriched the Introduction and discussion with more background about this topic. In the introduction, we added information about P influence on plants and more details about the P dynamic in soil. The P fertilizer consumption was added in the Introduction to explain the impact of high consumption and low-use efficiency in agriculture. More citations were added to the text and the reference list was updated.

Reviewer, Results: The results are very interesting BUT the data presentation needs more attention especially Figure 1,2, and 3) the resolution is very low the author is advised to provide NEW figures with better resolution.

Authors: The Figures with better resolution were added in the text based on your comments and other reviewers. Thanks for taking the time to read our manuscript.

Conclusion This section is well written and the conclusion is supported by the results of this study and please includes more data from the results, the most important findings. This section needs to include more application in different ways with a wider range for more crops in general. References, This section is well written and it is up To date.

Authors: The conclusion was edited based on your comments adding the highlights of stud, which were (i) the rate of 1.0 L/ton of microbial solution recorded the highest biological activity; (ii) the microbial solution improved the soil biological activity; (iii) the crop root growth was increased by microbial solution in both soil types; (iv) the plant phosphorus accumulation was increased by microbial solution. The reference list was updated

Round 2

Reviewer 1 Report

Dear Corresponding author

Congratulations for your paper. You modified your paper very well. Hope it will be published soon.

Regards